# Copy Number Changes and Allele Distribution Patterns of Chromosome 21 in B Cell Precursor Acute Lymphoblastic Leukemia

**DOI:** 10.3390/cancers13184597

**Published:** 2021-09-13

**Authors:** M. Reza Abbasi, Karin Nebral, Sabrina Haslinger, Andrea Inthal, Petra Zeitlhofer, Margit König, Dagmar Schinnerl, Stefan Köhrer, Sabine Strehl, Renate Panzer-Grümayer, Georg Mann, Andishe Attarbaschi, Oskar A. Haas

**Affiliations:** 1Labdia Labordiagnostik GmbH, Clinical Genetics, Zimmermannplatz 8, 1090 Vienna, Austria; rezaabbasi2@hotmail.com (M.R.A.); sabrina.haslinger@labdia.at (S.H.); andrea.inthal@labdia.at (A.I.); petra.zeitlhofer@labdia.at (P.Z.); margit.koenig@labdia.at (M.K.); stefan.koehrer@labdia.at (S.K.); rpanzerg@gmail.com (R.P.-G.); 2St. Anna Children’s Cancer Research Institute (CCRI), Clinical Genetics, Zimmermannplatz 10, 1090 Vienna, Austria; dagmar.schinnerl@ccri.at (D.S.); sabine.strehl@ccri.at (S.S.); drgeorgmann@gmail.com (G.M.); 3St. Anna Children’s Hospital, Pediatric Clinic, Medical University, Kinderspitalgasse 6, 1090 Vienna, Austria; andishe.attarbaschi@stanna.at

**Keywords:** childhood acute lymphoblastic leukemia, chromosome 21, Down syndrome, array analysis, short tandem repeats

## Abstract

**Simple Summary:**

Array analysis is an efficient method for defining in a single experiment all genome-wide large and fine-scale copy number abnormalities, as well as their corresponding allele patterns. Based on the results of the analysis that we performed in 578 children with acute lymphoblastic leukemia, we provide a comprehensive overview of the genetic subgroup-specific incidence and distribution of all the various types of chromosome 21 copy number alterations in this cohort, most of which are of eminent diagnostic and clinical relevance. By doing so, we also uncovered some unusual and difficult to explain discrepancies between copy number and allele distribution patterns that we investigated and eventually succeeded to resolve with polymorphic short tandem repeat analyses.

**Abstract:**

Chromosome 21 is the most affected chromosome in childhood acute lymphoblastic leukemia. Many of its numerical and structural abnormalities define diagnostically and clinically important subgroups. To obtain an overview about their types and their approximate genetic subgroup-specific incidence and distribution, we performed cytogenetic, FISH and array analyses in a total of 578 ALL patients (including 26 with a constitutional trisomy 21). The latter is the preferred method to assess genome-wide large and fine-scale copy number abnormalities (CNA) together with their corresponding allele distribution patterns. We identified a total of 258 cases (49%) with chromosome 21-associated CNA, a number that is perhaps lower-than-expected because *ETV6-RUNX1*-positive cases (11%) were significantly underrepresented in this array-analyzed cohort. Our most interesting observations relate to hyperdiploid leukemias with tetra- and pentasomies of chromosome 21 that develop in constitutionally trisomic patients. Utilizing comparative short tandem repeat analyses, we were able to prove that switches in the array-derived allele patterns are in fact meiotic recombination sites, which only become evident in patients with inborn trisomies that result from a meiosis 1 error. The detailed analysis of such cases may eventually provide important clues about the respective maldistribution mechanisms and the operative relevance of chromosome 21-specific regions in hyperdiploid leukemias.

## 1. Introduction

Chromosome 21 plays an extraordinary and unmatched role in the pathogenesis of childhood B cell precursor acute lymphoblastic leukemia (BCP ALL). A constitutional trisomy 21 is first of all the strongest leukemia predisposition factor that renders affected children susceptible to develop not only ALL but also transient myeloproliferative disorders and acute myeloblastic leukemia, especially that of the megakaryoblastic type [1,2,3,4,5,6,7]. Somatic gains of one or more chromosomes 21 are the most common abnormalities in sporadic ALL cases. Except for a subset of hyperdiploid cases, a trisomy 21 always evolves as a nonrandom secondary change in a genetic subgroup-restricted manner, for instance in those with an *ETV6-RUNX1* gene fusion [8,9,10,11,12] and those with a dicentric chromosome dic(9;20) [13,14]. Tetra- and occasionally also pentasomies of chromosomes 21, on the other hand, are typical hallmarks of a spectrum of aneuploid forms that comprise the classical hyperdiploid as well as various combinations of hyperhaploid and hypodiploid ones together with their corresponding hyperdiploid equivalents [15,16,17].

In addition to these pure numerical changes, chromosome 21 is also involved in structural rearrangements, most notably in the most common translocation in childhood ALL, the cryptic t(12;21)(p13;q13) with its resultant *ETV6-RUNX1* gene fusion [8,9,10,11,12]. The so-called intrachromosomal amplification of chromosome 21 (iAMP21), on the other hand, comprises regions of gains, amplifications, inversions and deletions that result from complex chromothripsis-like rearrangements [18,19,20]. Of note in this context is also the fact that individuals with an extremely rare constitutional Robertsonian translocation rob(15;21)(q10;q10) have a 2.700-fold increased risk to develop this particular type of leukemia [20].

Finally, one also needs to mention deletions of the *ERG* gene. As secondary but nearly exclusive markers, they typify 50 to 80% of CD371-positive, *DUX4*-rearranged cases [21,22,23] and, conversely, also serve as risk-dampening exclusion criteria for the heterogeneous group of *IKZF1*^plus^-positive leukemias, which are defined by deletions of the *IKZF1* gene together with those in various other B-cell developmental genes in the absence of *ERG* deletions [24].

Taken together, one can therefore estimate that chromosome 21 is involved in one way another in at least up to 65% of childhood BCP ALL cases [25].

Comparative genome hybridization/single nucleotide polymorphism (CGH/SNP) array analysis is the preferred method for the evaluation of genome-wide large and fine scale copy number deviations as well as allele distribution patterns [26,27,28,29]. Its integration into the routine diagnostic work-up of childhood ALL provides essential information that is nowadays particularly relevant for defining and delineating clinically important entities and treatment stratifications [16,24]. Such CGH/SNP arrays contain non-polymorphic oligos that detect quantitative changes, such as gains and losses of entire chromosomes as well as region specific deletions, duplications, and amplifications, together with SNP-containing ones that simultaneously reveal copy neutral losses of heterozygosity (CN-LOH) in form of complete or partial acquired uniparental disomies (UPD).

Based on large numbers of array-analyzed cases, we set out to obtain and provide an overview about the various types and approximate frequencies of partial and complete chromosome 21 CNA in various genetic subgroups, together with their corresponding allele distribution patterns, some of which may look conceivably obvious, whereas others provide some novel insights into their mechanistic origin, diagnostic meaning, and functional relevance. As we will specifically point out in more detail in some of the examples that we will present, the interpretation of unusual mosaic and/or complex copy number and associated allele pattern combinations can sometimes still be quite a challenging task.

## 2. Materials and Methods

### 2.1. Patients

Our analysis is based on the array analyses of a total of 622 samples that derived from 578 patients, the vast majority of whom were enrolled in treatment studies, including the ALL-BFM 90, ALL-BFM 95, ALL-BFM 2000, AIEOP-BFM ALL 2009, AIEOP-BFM ALL 2017, Interfant and EsPhALL.

Genetic subgroup classification was originally based on the results of cytogenetic, reverse-transcription (RT)-polymerase chain reaction (PCR) and fluorescence in situ hybridization (FISH) analyses, which were conducted in all cases with an extensive panel of ALL-specific probe sets that were continuously expanded and adapted according to the newly emerging diagnostic needs, and therefore consequently also supplemented with the array-derived information presented herein.

Since all these analyses were part of the diagnostic work-up they were covered by the respective study-related informed consents that were obtained from the patients and/or guardians in accordance with the Declaration of Helsinki and that were as such also approved by the local Institutional Review Boards.

### 2.2. Cytogenetic Analysis

Cytogenetic analyses were performed on cultured bone marrow cells that were prepared and banded according to standard methods.

### 2.3. Fluorescence In Situ Hybridization (FISH)

FISH analyses were performed with a set of commercially available dual color/dual fusion and dual color break apart probes according to the manufactures’ recommendations.

### 2.4. CGH/SNP Array Analysis

Genomic DNA was isolated from bone marrow mononuclear cells using the QIAamp^®^ DNA Blood Mini Kit (QIAGEN, Hilden, Germany). Array analysis was performed with the CytoScan™ HD array (2,670,000 markers including 750,000 single nucleotide polymorphisms (SNPs); Applied Biosystems™, Thermo Fisher Scientific, Waltham, MA, USA) according to the manufacturer’s instructions by commercial service providers. The obtained data were then analyzed in-house with the Chromosome Analysis Suite (ChAS; Applied Biosystems™, ThermoFisher Scientific) software package versions 3.0–4.1. All aberrations were mapped to the Genome Reference Consortium GRCh37, UCSC genome assembly hg19 reference genome. Genomic segments were filtered according to the following parameters: losses genome-wide ≥25 markers and for leukemia-associated genes ≥20 markers; gains genome-wide ≥50 markers and 50 kb size and for leukemia-associated genes ≥25 markers; LOH regions ≥50 markers and ≥3 Mb size. Copy number variants arising from B-cell and T-cell antigen receptor gene rearrangements as well as all known common benign CNVs were excluded.

### 2.5. Short Tandem Repeat (STR) Analysis

STR analysis was performed with the Aneufast™ Multiplex QF-PCR kit (aneufast.com; Genomed Ltd., Harrow, UK) according to the manufacturer instructions. The obtained QF-PCR products were analyzed by capillary electrophoresis on an ABI PRISM 3130XL Genetic Analyzer using ABI GeneMapper 5.0 Software by a commercial service provider. The Aneufast™ kit was originally designed for the rapid detection of aneuploidies of chromosomes 13, 18, 21, X and Y by quantitative fluorescent PCR (QF-PCR) and contains five STR markers for chromosome 21. QF-PCR amplification of STR markers generates a fluorescent product that is directly proportional to the amount of the target sequence present in the initial template. The peak height is a measure of fluorescent activity and therefore directly proportional to the amount of the fluorescent products. For each marker, peak ratios are calculated from the results of the capillary electrophoresis. Normal heterozygous individuals have an approximately equal representation of each allele and will therefore have a peak ratio of 1:1. Homozygous individuals with only one peak at a given marker are not informative for that marker. Trisomic cases either carry three different alleles with equal intensities or two alleles with unbalanced ratios (1:2 or 2:1).

### 2.6. RNA Sequencing (RNAseq)

RNAseq was performed by a commercial service provider (Next-Generation Sequencing Unit, Vienna BioCenter Core Facilities, VBCF; www.viennabiocenter.org/facilities 4 September 2021.) in a selected number of B-other cases as previously described in detail [22].

## 3. Results and Discussion

Table 1 provides a global overview of the chromosome copy number and allele distribution patterns of chromosome 21, which we identified in a total of 622 array-analyzed samples from 578 patients. They comprise 565 diagnostic and 57 relapse ones, 40 of which were matched pairs. Included are also 26 diagnostic and six relapse samples from 26 patients with a constitutional trisomy 21.

Those 304 cases that were diagnosed between May 2015 and the end of 2019 were analyzed in a consecutive prospective manner, whereas the remaining 274 ones derive from various retrospective studies and primarily included aneuploid and specific B-other subgroups as well as Down syndrome cases but hardly any *ETV6-RUNX1*-positive ones. This explains the biased low number of *ETV6-RUNX1*-positive cases (66/578; 11%) and consequently also the overall comparatively low frequency of chromosome 21 alterations (258/578; 49%) in our cohort. The results presented herein can therefore only provide an approximate but nevertheless representative estimate of the frequency and type of chromosome 21 abnormalities in the various entities, which is, however, sufficient for the intended purpose of our study.

### 3.1. Disomy 21

Out of 552 disomic cases, 215 retained a regular pair of heterozygous chromosomes 21 in their leukemic clone, an example of which is shown in Figure 1a. They include 14 with matched diagnostic/relapse samples and one case with two relapses. Chromosome 21 abnormalities neither emerged in *KMT2A* nor in *TCF3-PBX1* rearranged cases, either because they are unnecessary and would not provide any further benefit or, perhaps even more likely, because they would even interfere with and preclude the respective disease process. Except for three pure hypodiploid cases, these heterozygous disomic cases are therefore not further relevant for the specific purpose of our study.

Two disomic B-other patients, one of whom relapsed, acquired a CN-LOH (Figure 1b). The small centromeric heterozygotic region that was retained in both cases confirms that this abnormality resulted from a mitotic recombination between two originally heterozygous chromosomes 21 rather than from the bi-chromatid mis-segregation of a single homologue [30,31]. Such acquired CN-LOHs commonly designate the duplication of either inborn or acquired original heterozygous mutations in disease-relevant genes that are contained within this region. We therefore sequenced the most likely candidate, the *RUNX1* gene, in both patients and indeed identified an acquired homozygous c.637C>T (p.Gln213Ter) mutation in exon 7 in one of them (case 1). She had a normal karyotype and no other array-verifiable CNAs, but another small CN-LOH at chromosome (9)(q34.11q34.2). The only obvious karyotype abnormality in case 2 was a t(15;16)(q14;q22). Subsequent RNA-Seq and RT-PCR analyses revealed that this translocation concurred with a 5′-*ZFHX3*/3′-*SLC12A6* gene fusion and array-detectable small, rearrangement-associated deletions of both genes. The *ZFHX3* gene encodes the alpha-fetoprotein enhancer-binding protein and the *SLC12A6* gene the potassium chloride cotransporter 3 (genecards.org). In addition to the breakpoint-associated deletions of both fusion gene partners, we also found a homozygous deletion of the *CDKN2A* and heterozygous ones of the *CDKN2B* and the *TBL1XR1* genes.

### 3.2. CNA in ETV6-RUNX1-Positive Cases

The array-based recognizable characteristic features of *ETV6-RUNX1*-positive cases are genome-wide heterogeneous yet typical deletions that result from illegitimate *RAG1/RAG2*-mediated V-(D)-J recombination processes and target primarily genes that encode crucial components of B-cell development and normal hematopoiesis [8,9,32,33]. As regards chromosome 21, in particular, there are four distinct types of CNAs that comprise direct fusion-related intragenic *RUNX1* deletions, and, as secondary changes, the duplication of the non-rearranged homologue as well as duplications of either the distal or the proximal translocated parts in form of a der(12) or a der(21), respectively. In our cohort of 66 *ETV6-RUNX1*-positive cases, we identified 12 with an extra chromosome 21, one with a breakpoint-related *RUNX1* intron 2 deletion, four with two der(12) copies and one with two der(21) copies (Figure 2 and Figure 3 and Figure A1). Previous studies, which were all based on similar small numbers of *ETV6-RUNX1*-positive patients and analyzed with cytogenetics, various types and combinations of FISH probes and/or low-resolution SNP arrays, reported a trisomy 21 in up to 23%, a der(21) in up to 15% and a der(12) in approximately 8% of cases [8,9,12,32,33,34,35].

Chromosome analyses are virtually unable to delineate normal chromosomes 21 and 12 from der(21) and der(12) ones, respectively. Moreover, with the commonly used dual color/dual fusion interphase FISH approach alone one is similar unable to interpret the origin of supernumerary fusion products. Even with FISH, an unequivocal distinction between a der(21) and a der(12) becomes only feasible either on metaphases or, in interphases, only with the additional application of dual color *ETV6*-5′ and 3′ break apart probes (Figure A1) and/or with array analyses. It is worth noting that all these secondary duplications are mutually exclusive, which can be viewed as indirect, albeit resounding evidence that both centromeric and telomeric parts of the *RUNX1* gene that border the breakpoint may be either equally relevant for or at least not restraining the disease process if neither of them exceeds three copies.

### 3.3. CNA and Allele Patterns in iAMP21 Cases

We identified 19 cases with an iAMP21, 17 of which were analyzed at diagnosis, two only at relapse and five on both occasions. The matched samples of these cases confirmed that, once formed, these characteristic alterations remain remarkable stable over the entire course of the disease (Figure 4).

The shared pathognomonic hallmark of this entity is a recurrent amplification of the RUNX1-containing region that represents the relevant diagnostic part of otherwise chromosome 21-wide chromothripsis-like complex rearrangements. Although array analyses reveal characteristic patterns of the lost and gained parts as well as good evidence that the respective rearrangements involve a single allele, only metaphase FISH analysis alone can secure that these alterations are indeed located on a single homolog [18,19,20]. Recent mouse experiments indicate that this outstanding amplification is most likely also the essential disease-initiating event, because the germline triplication of 31 orthologous genes, which overlap with this specific region, is sufficient for transforming mouse progenitor B cells and that the overexpression of *HMGN1* alone can promote both B cell proliferation in vitro and B-ALL development in vivo [36,37].

### 3.4. ERG Deletions

Recurrent intragenic *ERG* deletions that result from illegitimate *RAG1/RAG2*-mediated V-(D)-J recombination processes are always secondary changes that occur in nearly all *DUX4*-rearranged cases [21,22,23,38,39]. On the other hand, they also serve as a risk-reducing exclusion criterion for *IKZF1^plus^*-positive cases [24]. We came across 24 such B-other cases (including one with a matched relapse), whose crucial diagnostic features were eleven distinct types of purely monoallelic alterations (Figure 5). *ERG*-deletions were seen in 21 CD371-positive and RNAseq-verified DUX4-rearranged cases. The respective information was not available in one case, one of the two CD371-negative ones would have otherwise qualified as *IKZF1^plus^* and the other one had only a del(9)(p13.2) that encompassed the *PAX5* gene.

Although array analyses are clearly sufficient to define the extent and, consequently, also the type of such deletions, they still lack the sensitivity of PCR-based methods. With those it was shown that *ERG* deletions are not only heterogeneous but also predominantly polyclonal and the presence of even small subclones may apparently already be of prognostic relevance [21]. Irrespective of the functional consequences for the affected cells themselves, this finding is still quite perplexing, because as passenger mutations such deletions are merely the unfortunate by-product of a more extensive collateral DNA damage [21]. If confirmed, such *ERG* deletions could thus exert a kind of “latent or mini driver” effect, a situation that is somehow reminiscent of the one also observed in leukemias with small *P2RY8-CRLF2*-positive subclones [40,41,42,43].

### 3.5. Constitutional Trisomy 21

In our cohort, we had 26 cases with Down syndrome, 25 of which were studied at diagnosis and one only at relapse. Overall, there were five cases with six relapses, since one of the four matched ones relapsed twice. Trisomy 21 remained unaltered in the leukemic cells of 19 cases (one *ETV6-RUNX1*-positive, one hyperdiploid and 17 B-other ones). Of these, one hyperdiploid and two B-other cases were studied both at diagnosis and relapse. Nine of the 12 *CRLF2*-overexpressing B-other cases had a *P2RY8-CRLF2* and three an *IGH-CRLF2* fusion. The seven remaining Down syndrome cases had hyperdiploid leukemias, three with four and four with five copies of chromosomes 21, respectively.

A constitutional trisomy 21 can either result from a meiosis 1, meiosis 2 or postzygotic mitotic nondisjunction error. The vast majority of the extra copies are of maternal origin and even those that seem to arise at meiosis 2 can in fact be traced back to problems that had occurred already at meiosis 1 [44,45,46]. Approximately 10% of such trisomies result from a paternal meiotic nondisjunction and only a very small proportion of approximately 2% are attributable to post-zygotic mitotic nondisjunction failures [44].

Meiosis 1 errors can be discriminated from meiosis 2 or mitotic ones by determining the allelic composition of the three chromosomes with polymorphic markers, for instance with highly variable short tandem repeats (STR) that are dispersed over the entire chromosome [45]. The failure to separate an at least partially heterozygous pair of chromosome 21 in meiosis 1 will then create a regional “1 + 1 + 1” pattern, whereas the co-segregation of both sister chromatids of a single homologue in meiosis 2 or mitosis can only generate a “2 + 1” pattern [44,46,47]. One fundamental drawback of the array technology is that, for technical reasons, one cannot determine which SNPs are located on which alleles. This renders it impossible to clarify the phase of origin as well as the parental provenience of the surplus chromosome with arrays (alone) as well as to identify the presence or location of potential recombination sites. There are only three conditions in which either a switch of or an unequal distribution of heterozygous alleles will expose them [28]. The first one concerns mitotic recombination that, as shown in Figure 1, for instance, will turn a disomic heterozygous region into a homozygous one and thereby produce a discernable CN-LOH of the affected segment [30,31,48]. Meiotic recombination sites, on the other hand, will only become apparent in specific forms of mosaicism, for instance in leukemias with tetrasomies and pentasomies in constitutional trisomic cases with three discernable chromosome 21 homologues. Lastly, an admixture of cells from two blood related individuals, as can be found either in natural occurring or iatrogenic transplant-associated forms of chimerism that involve either parents or siblings as donors and recipients, respectively, will reveal the genome-wide dispersal of such meiotic recombination sites [48].

### 3.6. Acquired Trisomy 21

Except for hyperdiploid cases, a single extra copy of chromosome 21 is always a nonrandom secondary event that only emerges in leukemias with certain primary changes (Table 1).

We identified a third chromosome 21 in 69 samples of 64 cases, 61 of which were studied at diagnosis and three only at relapse. In one of the four matched diagnosis/relapse cases disease recurred twice. Apart from the above-mentioned 12 *ETV6-RUNX1*-positive cases (one case with two relapses), a somatic gain of chromosome 21 was seen in one *BCR-ABL1*-positive, 20 hyperdiploid and 31 B-other (3 relapses, 3 matched) cases.

Amongst other changes, the only *BCR-ABL1*-positive case also contained a trisomy 21 at diagnosis and a tetrasomy 21 at relapse, respectively, where one of the four homologues was also affected by a 276 kb deletion. Further details of the respective changes can be found in Figure A2.

In keeping with the literature, we found a trisomy 21 in approximately 12% of our hyperdiploid cases [49]. Whether in these instances the acquisition of an extra chromosome 21 is a direct consequence of the original nondisjunction event or whether it is caused by a secondary, single copy loss from a primary tetrasomy, is virtually impossible to determine and must therefore remain elusive.

Nineteen of the 48 trisomic B-other cases occurred together with a *CRLF2*-overexpressing rearrangement, one of which had also a dic(9;20) and was therefore assigned to this particular entity [40,43]. Conversely, 19 (50%) of the total 38 *CRLF2*-positive cases had either a constitutional or acquired trisomy 21, a finding that underlines the close association and interdependence of these two alterations and confirms that trisomy 21 is not only an important predisposing but, in the acquired form, also a prevalent survival and progression factor in this specific leukemia subset.

Perhaps to a slightly lesser extent, the latter is also true for cases with a dic(9;20), since 10 of our 20 cases had an extra copy of chromosome 21 [13,14]. In one case, array analysis uncovered an intriguing discrepancy, namely the presence of three copies together with an STR-verified evenly balanced heterozygous allele pattern (Figure 6; designated in Table 1 with ***). We were unfortunately not able to satisfactorily resolve this perplexing combination, which in our opinion can only be explained by postulating the presence of two distinct, approximately equally sized trisomic clones, one with an AAB and one with an ABB allele pattern. Whether this is indeed the case can only be resolved with single cell sequencing.

As shown recently by Gu et al., an acquired trisomy 21 may also serve as an important indicator and surrogate marker for another rare B-ALL subtype that is defined by a distinct gene expression profile and an *IKZF1* N159Y (p.Asn159Tyr) missense mutation [50]. The fact that six of the eight reported cases had and extra copy chromosome 21 but no other focal CNAs or recurrent sequence mutations attests the crucial contribution of chromosomes 21 in the development of this specific subtype.

After excluding all otherwise classifiable B-other subtypes with a solitary trisomy 21 in our cohort, six cases remained that were most likely candidates for being members of this specific subtype and, indeed, we identified the respective expression profile and *IKZF1* N159Y mutation in three of the four investigated cases (Table 2).

### 3.7. Tetrasomy 21

A tetrasomy 21 is essentially the defining feature and diagnostic hallmark of hyperdiploid ALL forms [17,51]. The different allele distribution patterns that underlie the various kinds of tetrasomy in constitutional disomic and trisomic individuals are schematically explained in Figure 7. Representative examples of the ensuing array patterns are shown in Figure 8 and Figure 9.

We ascertained a total of 165 such cases (163 at diagnosis and two at relapse only), 162 in constitutional disomic and three in constitutional trisomic cases. Ten of them were seen both at diagnosis and relapse, two of which had two relapses. One constitutional disomic case had a trisomy at diagnosis and a “3 + 1” tetrasomy at relapse. There were 154 “classical” hyperdiploid cases, 145 of which had a “2 + 2”, eight a “3 + 1” and one a “2 + 1 + 1” allele distribution pattern (Figure 8 and Figure 9). In addition, there were seven cases that had either a hyperhaploid and/or a hyperdiploid clone with a genome-wide CN-LOH, which means that the respective hyperhaploid clones contained a biparental pair and the corresponding hyperdiploid versions a biparental tetrasomy of chromosome 21 [51]. Low hyperdiploid genome-wide CN-LOH cases form a unique array-definable entity. They have only between 48 and up to 54 chromosomes and most likely derive from a single mitotic cell in a single step rather than from the genomic duplication of an afore generated hyperhaploid cell clone [51].

One of the pure hyperdiploid cases was found to be *BCR-ABL1*-positive in the RT-PCR analysis. Subsequent validation with a dual-color/dual-fusion FISH probe revealed only a single fusion signal in 11% of the respective interphase cells, that was most likely located on the der(22). We are only aware of two similar adult *BCR-ABL1*-positive cases in which this fusion apparently arose as a secondary change in hyperhaploid/hyperdiploid leukemias [52,53].

### 3.8. Pentasomy 21

Representative examples of the different allele distribution patterns in acquired pentasomies of constitutionally disomic as well as trisomic patients are schematically explained in Figure 7 and the respective array images are shown in Figure 8 and Figure 10.

Overall, array analysis discovered 12 patients with a pentasomy 21. Irrespective of whether they were constitutional disomic (eight cases) or trisomic (four cases), they had all the expectable “3 + 2” allele pattern. In all but one disomic B-other case, which was assigned to this category because of its near-triploid karyotype, pentasomy 21 was always part of a hyperdiploid genome. Copy number validation with a *RUNX1* FISH probe also uncovered small subclones with three and four chromosomes 21 in all of them. Whether these resulted from chromosome loss or, *vice versa*, whether pentasomy arose from successive gains remains to be seen, although one may concede that for a cell the loss of a superfluous chromosome is generally easier to cope with than acquiring an additional one [51].

## 4. Conclusions

Our comprehensive overview provides illustrative examples of all the various array-detectable chromosome 21 abnormalities and their specific role in various subtypes of childhood B-ALL. The presence of an extra copy of chromosome 21 strongly influences the manifestation and maintenance of specific types of leukemias, in particular those with a *CRLF2*-overexpressing lesion and those with a dic(9;20). Whereas an inborn trisomy 21 primarily predisposes to the development of the former, it does not at all in the later, although the acquisition of an extra copy of chromosome 21 is otherwise a very common attribute of both leukemia types in constitutional normal individuals. Moreover, as a solitary change, an acquired trisomy 21 can serve as an important indicator and surrogate marker for the recently delineated rare subtype with a distinct expression profile and an *IKZF1* N159Y mutation. Since arrays will only highlight the overall most prominent changes that are present in the analyzed bulk DNA, we consider it of utmost importance to validate any conspicuous CNA result or discrepant CNA/allele configuration with FISH and the corresponding allele composition with STR analyses (or more sophisticated techniques), which will also suffice to obtain a more refined insight into the clonal composition of the respective cell populations. By doing so, we were able to authenticate the preserved heterozygous allele pattern of a trisomy 21 in a dic(9;20) case and clarify the origin of those unique intrachromosomal allele changes in hyperdiploid leukemias with tetra- and pentasomies that develop in constitutional trisomic patients. We were able to show that such visible allele switches uncover meiotic recombination sites that are created by a copy number imbalance of three distinct alleles, which attests that the patient’s inborn trisomy results from a meiosis 1 nondisjunction error. A more wide-spread and detailed analysis of such intriguing cases may eventually provide important clues, not only about the respective maldistribution mechanisms as such, but also whether such recombination events might be (co-)responsible for creating regions on chromosome 21 that either increase the likelihood to develop hyperdiploid leukemias or may be otherwise operational relevant in the transformation process.

## Figures and Tables

**Figure 1 cancers-13-04597-f001:**
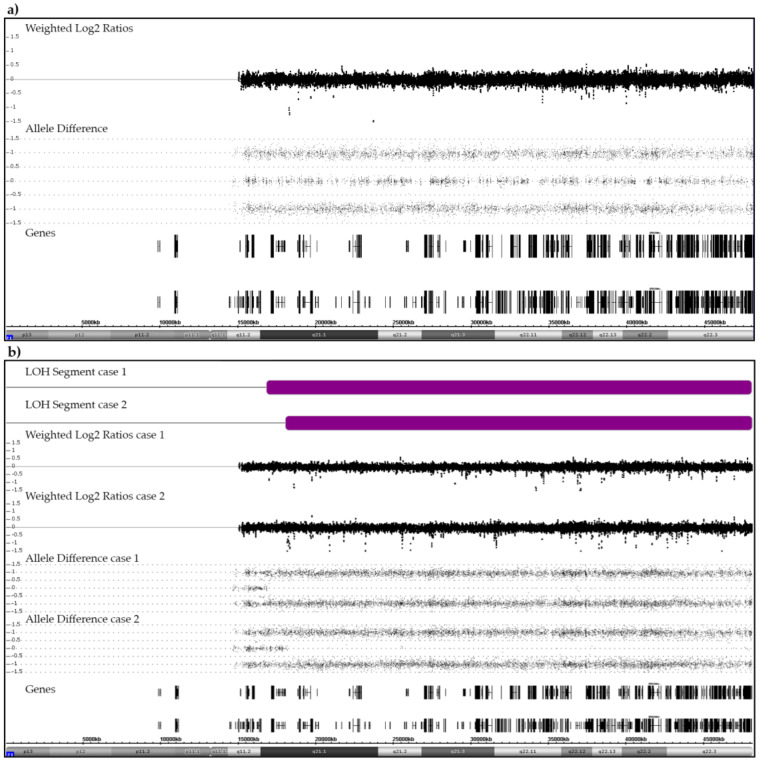
Examples of SNP/CGH array patterns that were obtained from a case with a normal heterozygous (**a**) and from two cases with a partially homozygous pair of chromosomes 21 (**b**). (**a**) The allele distribution pattern of a normal pair of heterozygous chromosomes 21 serves as reference point for the interpretation of any deviant allele distribution patterns discussed herein. The weighted log2 ratios and the SNP allele pattern are shown in the middle part. Black vertical and horizontal bars in the bottom part indicate the location of genes on chromosome 21. (**b**) Two cases with a CN-LOH (purple bars). In case 1, the size of the CN-LOH is 31.3 Mb (21q21.1q22.3; chr21:16,818,166–48,084,820). The recombination took place between the *NRIP1* and *USP25* genes (chr21:16,437,126–17,102,253). In case 2, the size of the CN-LOH is 30 Mb (21q21.1q22.3; chr21:18,031,461–48,084,820). The recombination took place between the *MIR99AHG* (NR_136542) and *LINC01549* (NR_037585) genes (chr21: 17,983,094–18,811,208).

**Figure 2 cancers-13-04597-f002:**
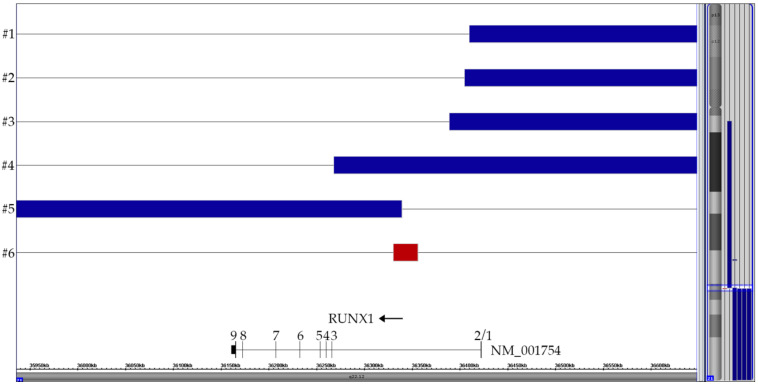
Summary of the six partial chromosome 21 CNAs that are all the direct or indirect byproduct of the t(12;21) with its *ETV6-RUNX1* gene fusion, respectively. Only the relevant part surrounding the *RUNX1* breakpoint in intron 2 is shown. The horizontal top four blue lanes represent cases with two der(12) copies, the bottom blue one the case with a duplicated der(21) and the red one the case with the small breakpoint-associated 25 kb deletion (72 markers). The corresponding vertical lines on the right of the ideogram indicated the respective extent of these CNA on chromosome 21.

**Figure 3 cancers-13-04597-f003:**
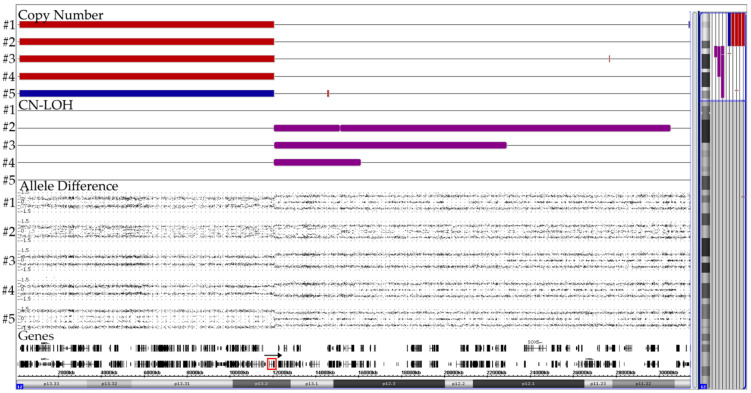
Summary of the CNA patterns on chromosome 12 in four cases with two non-identical der(12)t(12;21) (#1–4) copies and one with a der(21)t(12;21) (#5). The corresponding case numbers are shown on the left and the location of the *ETV6* gene is marked with a red box in the gene track. The rearrangements in the *ETV6*-surrounding region are far more complex than those in the *RUNX1*-surrounding one and essentially reflect the difficulties that accompany the successful fusion of the reversely orientated *RUNX1* and *ETV6* genes. In the four cases with a der(12), the telomeric part (including *ETV6*) of the second chromosome 12 is deleted. The CN-LOH regions of the adjacent extended *ETV6*-containing centromeric part of both homologues in three of them provide convincing evidence that these complex configurations must be produced simultaneously in a single step [32].

**Figure 4 cancers-13-04597-f004:**
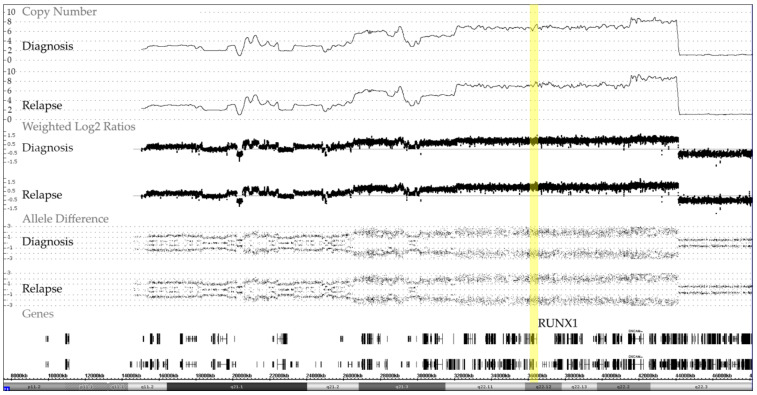
Representative example of chromosome 21 CNA and allele distribution patterns in matched diagnosis and relapse samples of a case with an iAMP21, which also illustrates the remarkable stability of the complex chromosome-wide rearrangement. The common essential feature of all iAMP21 cases is the high-level amplification of the *RUNX1*-containing region (yellow bar) that spans about 5.1 Mb and ranges from approximately 32.8 to 37.9 Mb and the virtually always co-occurring deletion of the distal telomeric part [18,19,20].

**Figure 5 cancers-13-04597-f005:**
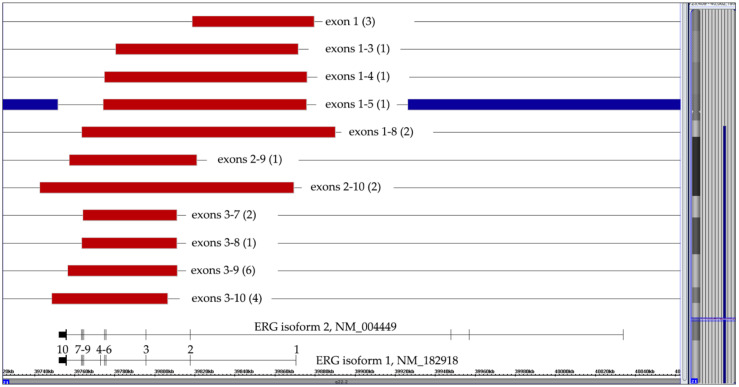
Summary of 24 cases with eleven different types of isoform 1-projected *ERG* deletions. The numbers of the respective cases are in brackets. Since the exon 1–5 deletion in the case with a trisomy 21 (represented by the blue lines) affected two alleles, the deletion must have already taken place before the chromosome duplicated.

**Figure 6 cancers-13-04597-f006:**
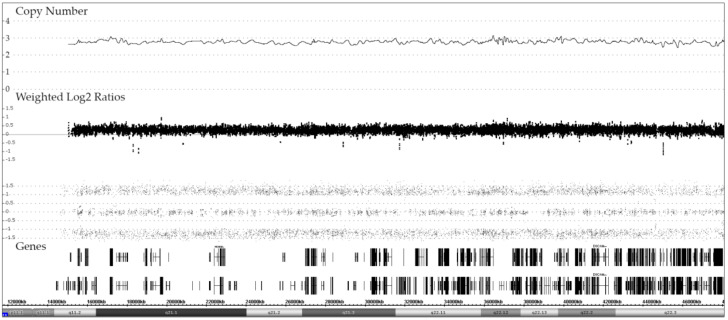
Array image of a unique dic(9;20) case with three chromosome 21 copies and an evenly balanced heterozygous allele pattern. The presence of the trisomy was verified with FISH and the biallelic allele pattern with STR analyses (data not shown). The karyotype of the leukemic cell population is 46,XX,del(2)(p23?)[3],dic(9;20)(p13;q11),+21 and the array one 46,XX,del(9)(p24.3p13.1),del(9)(p21.3)x2,del(20)(q11.21q13.33)/dic(9;20),+21, respectively.

**Figure 7 cancers-13-04597-f007:**
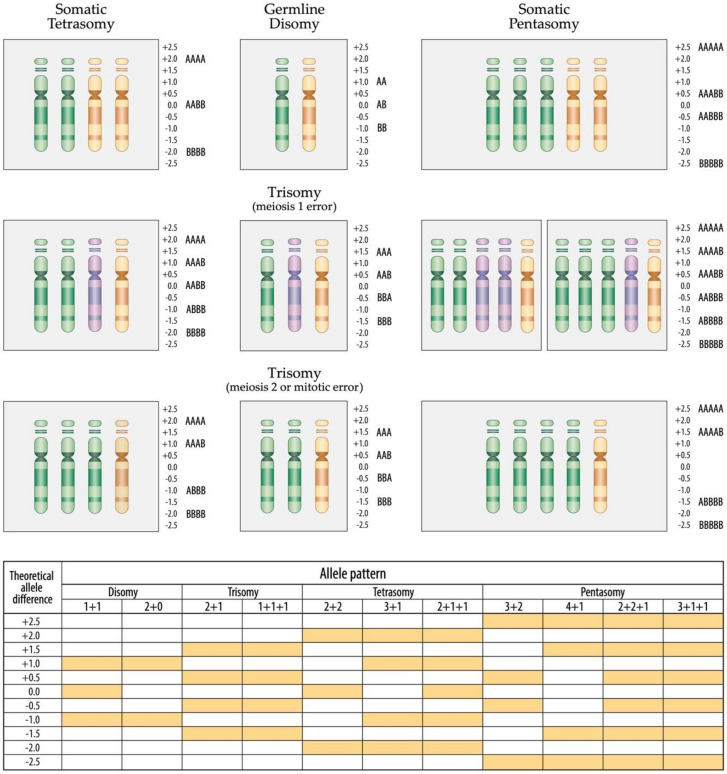
Schematic overview of chromosome 21 homologue and corresponding allele distribution patterns in tetra- and pentasomic hyperdiploid leukemias in constitutionally disomic and trisomic cases, respectively. Only one example of the various possible permutations of homologue distributions is shown in each instance. All variants of tetra- and pentasomies in constitutional disomic or trisomic individuals that result from a meiosis 2 or mitotic error, produce identical allele distribution patterns, in case of tetrasomies either a "2 + 2” or “3 + 1” and in case of pentasomies a "3 + 2” or “4 + 1” one. By far the vast majority of tetrasomies derive from the duplication of both parental homologues (“2 + 2”) in both constitutional disomic as well as trisomic cases, whereas the alternative “3 + 1” pattern is much rarer [17,51]. The same applies also to pentasomies: a “3 + 2” is significantly more common than a “4 + 1” pattern. Of particular interest are those tetra- and pentasomic leukemias that indeed contain three distinct alleles, which can only be found in constitutional trisomic individuals who emerge from a meiosis 1 error. Such tetrasomies can thus only have a “2 + 1 + 1” and pentasomies only a “2 + 2 + 1” or 3 + 1 + 1 combination of homologues, respectively. The scale of the allele difference on the side of the ideograms as well as the one displayed in the schematic layout on the bottom of the figure can only be taken as a mathematically determined optimum. As becomes especially evident in all the array images that derive from increased copy number alterations, such a theoretical optimum can in practice never be achieved. In truth, one has always to keep in mind that the allele difference as well as the Log2 ratio will never reach the mathematically expected amplitude, because its height will always depend on and be compressed by a variety of factors, including the technical quality of the hybridization, the applied computational algorithm, the respective copy number, the clonal composition of the investigated sample as well as the respective genotype configuration. The respective allele distribution pattern together with the obtained copy number per se, in contrast, delivers a much more accurate and reliable information (http://assets.thermofisher.com/TFS-Assets/GSD/Technical-Notes/chas-4–1-user-guide.pdf accessed on 11 February 2020).

**Figure 8 cancers-13-04597-f008:**
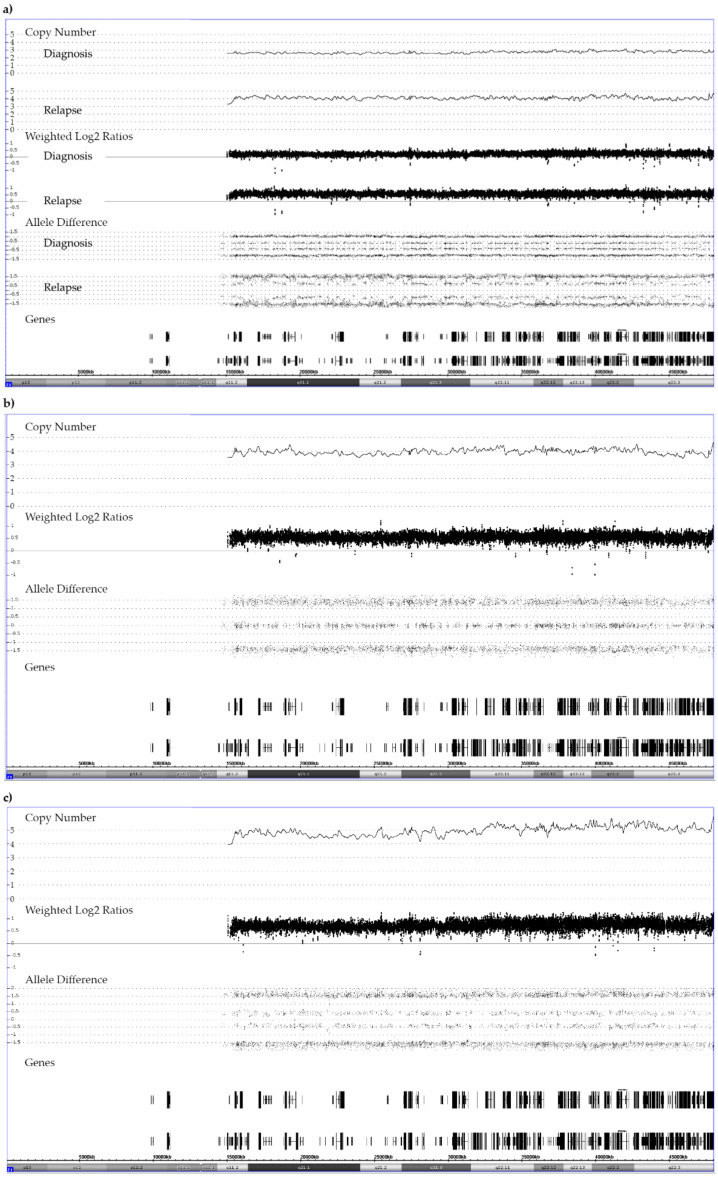
Representative examples of the allele distribution patterns in constitutionally disomic patients with (**a**) a “2 + 1” trisomy at diagnosis and a “3 + 1 tetrasomy” at relapse, (**b**) a “2 + 2” tetrasomy and (**c**) a “3 + 2” pentasomy of chromosome 21. The corresponding *RUNX1* FISH-based enumeration of chromosome 21 copies revealed the following clonal distribution: (**a**) at diagnosis 19% with two, 81% with three and at relapse 19% with two, 9% with three, 63% with four and 9% with five copies; (**b**) 7% with two, 13% with three, 80% with four copies; (**c**) 6% with two, 6% with four, 88% with five copies.

**Figure 9 cancers-13-04597-f009:**
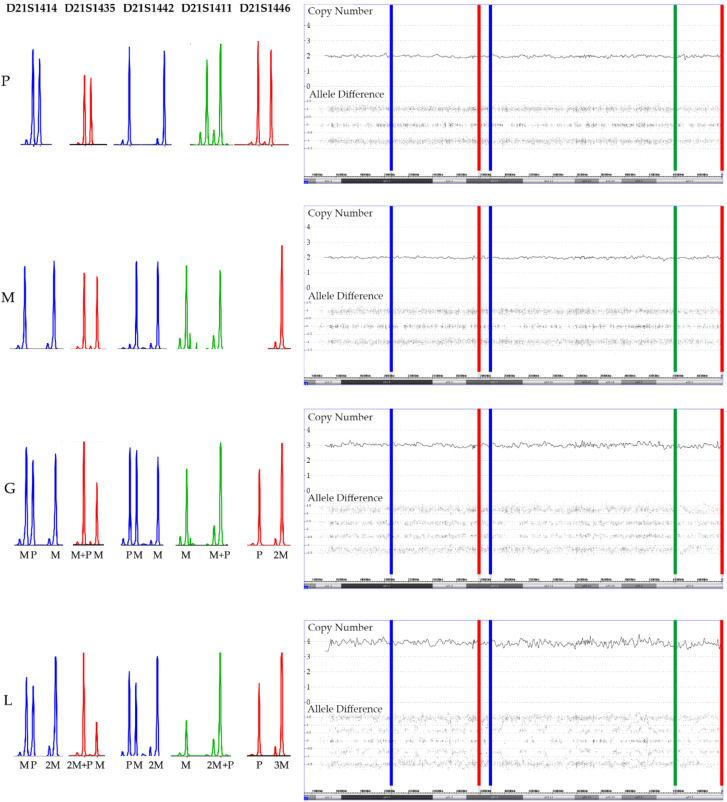
Add-on analyses of five short tandem repeat markers (STR) clarifies the parental origin (M: maternal, P: paternal) of the array-secured surplus chromosomes in the trisomic germline (G) and the tetrasomic leukemic cells (L) in a patient with a constitutional trisomy 21. The location of the five STR markers is marked in the array image in successive order. As evidenced by two peaks of a similar height, all paternal and four maternal markers are heterozygous and only the maternal D21S1446 marker is homozygous (only one but higher peak). Two of the markers are informative (D21S1414 and D21S1442) and clearly prove the maternal meiosis 1 origin of the extra chromosomes and, although it cannot be decided whether the three non-informative ones (D21S1435, D21S1411 and D21S1446) are uni- or biparental derived, their homozygosity is at least also compatible with this interpretation. The STR pattern of the leukemic cells not only prove that one of the maternal homologues is duplicated but also uncovers which of the heterozygous D21S1414, D21S1442, D21S1411 and D21S1435 alleles belong together and define it. What also becomes clear from this comparative analysis is that trisomies produce the same array allele distribution patterns, irrespective of whether they are composed of two (2 + 1) or three (1 + 1 + 1) distinct alleles. Copy number validation with a *RUNX1* FISH probe unveiled that 25% of the leukemic cells had three and 75% four chromosomes 21.

**Figure 10 cancers-13-04597-f010:**
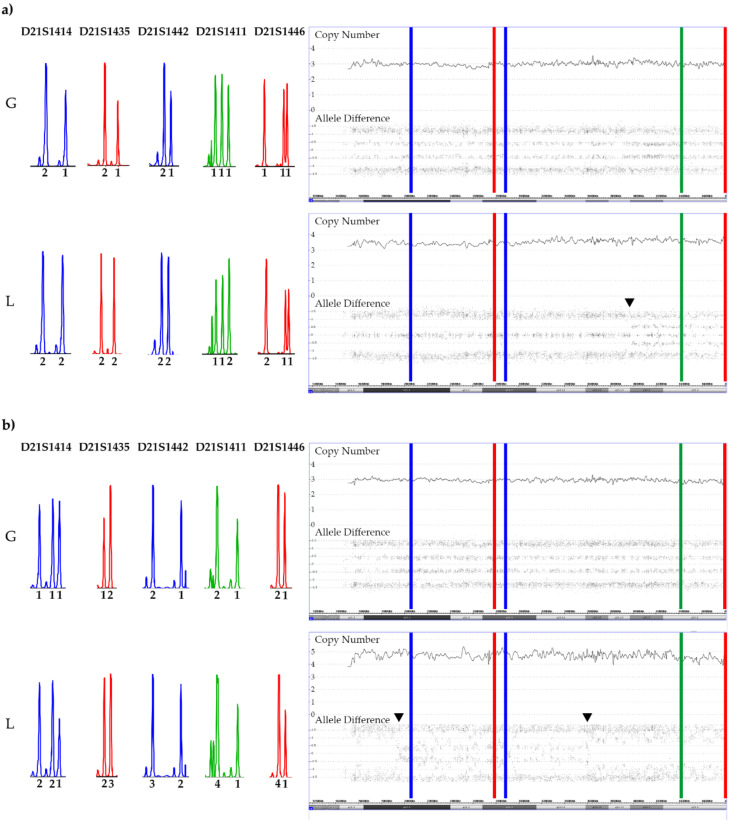
SNP array and STR patterns of the leukemic and remission sample of constitutional trisomic patients with an acquired tetrasomy (**a**) or pentasomy (**b**) of chromosome 21. The corresponding allele copy numbers are indicated below the respective STR markers. As evidenced by the STR markers with three alleles, both cases result from a (most likely maternal) meiosis 1 error. However, without parental material we were unfortunately not able to unequivocally define which parental homologues were indeed gained in the tetra- and pentasomic clones, respectively. In the tetrasomic case (**a**), the first four STR markers on the right and the fifth one on the left define the duplicated chromosome. Moreover, in this example a meiosis 1-associated switch from a “2 + 2” to a “2 + 1 + 1” configuration uncovered a recombination site that is clearly visible in the array-derived allele distribution pattern, and which is incidentally located in the ERG gene (indicated by the black arrow). Copy number validation with a *RUNX1* FISH probe identified a trisomy in 21% and a tetrasomy in 79% of the leukemic cells. The situation is similar in the pentasomic case (**b**), in which two of the four STR markers with homozygous alleles (“2 + 1”; D21S1435, D21S1442, D21S1411and D21S1446), which might have been either uniparental or biparental-derived, became duplicated (D21S1411and D21S1446) together with two of the three D21S1414 and one each of the D21S1435, D21S1442 markers. There are two meiotic recombination sites, which in this instance resulted from a “4 + 1” to a”2 + 3” and again a “4 + 1” allele distribution switch (indicated by the black arrows). Copy number validation with a *RUNX1* FISH probe identified a trisomy in 29% and a difficult to discriminate tetra- or pentasomy in 71% of the leukemic cells. We found such recombination sites in two of the three tetrasomic and in two of the four pentasomic cases.

**Table 1 cancers-13-04597-t001:** Summary of all the types of copy number abnormalities and allele distribution patterns that we identified in 578 children with various ALL subtypes.

Genetic Subtype	Patients	Samples	Male	Female	Diagnosis	Relapse	Matched	Down	Copy Number & Allele Distribution							
									2		3	4			5	6
									1 + 1	1 *	2 + 1	2 + 2	3 + 1	2 + 1+1	3 + 2	3 + 3
*ETV6-RUNX1* fusion	66	69	34	32	66	3	2	1	54	0	12	0	0	0	0	0
*KMT2A* fusions	17	21	6	11	17	4	3	0	17	0	0	0	0	0	0	0
*BCR-ABL1* fusion	8	10	7	1	6	4	2	0	7	0	1	0	0	0	0	0
*TCF3-PBX1* fusion	16	16	8	8	15	1	0	0	16	0	0	0	0	0	0	0
Aneuploid	200	212	115	85	197	15	11	8	3	0	21	156	8	1	11	0
Hyperdiploid, classical	186	198	106	80	184	14	11	8	0	0	21	145	8	1	11	0
Hyperdiploid, GW CN-LOH **	5	5	2	3	5	0	0	0	0	0	0	5	0	0	0	0
Hyperhaploid/hyperdiploid	2	2	1	1	2	0	0	0	0	0	0	2	0	0	0	0
Hypodiploid only	3	3	3	0	3	0	0	0	3	0	0	0	0	0	0	0
Hypodiploid/hyperdiploid	4	4	3	1	3	1	0	0	0	0	0	4	0	0	0	0
B-other	271	294	157	114	264	30	22	17	196	2	48	5	0	0	1	0
iAMP21	19	24	7	12	17	7	5	0								
*P2RY8-CRLF2* fusion	26	30	18	8	26	4	3	9	10	0	15	1	0	0	0	0
*IGH-CRLF2* fusion	12	12	5	7	11	1	0	3	8	0	4	0	0	0	0	0
dic(9;20) ***	20	24	9	11	20	4	4	0	10	0	10	0	0	0	0	0
i(9q)	13	13	11	2	13	0	0	2	11	0	2	0	0	0	0	0
*ERG* deletions	24	25	15	10	24	1	0	0	24	0	0	0	0	0	0	0
Not further specified	157	166	92	64	153	13	10	3	133	2	17	4	0	0	1	0
Total	578	622	327	251	565	57	40	26	293	2	82	161	8	1	12	0

GW CN-LOH, genome-wide copy neutral loss of heterozygosity; * nearly complete CN-LOH of the entire chromosome; ** exclusively homozygous disomies and biparental-derived heterozygous tetrasomies; *** includes one case with trisomy 21 that had an evenly balanced heterozygous allele pattern and one case with a *P2RY8-CRLF2* fusion.

**Table 2 cancers-13-04597-t002:** Array patterns of six B-other cases with a solitary trisomy 21. Three of the four investigated cases had an *IKZF1* N159Y mutation and the associated distinct expression pattern. The mutation was validated with PCR and the following primers: *IKZF1* intron5-F1: AAGGAGCTGGCAGGTTTAGTC and *IKZF1* intron6-R2: GGTTAGCCAGCAAGGACACA.

Patient	*IKZF1* N159Y	RNA seq Profile	Array Pattern
1	no	no	47, XY, del(9) (p13.2)/PAX5,+ 21
2	yes	yes	47, XY, del(2)(p12p11.2),del(16)(p11.2),+ 21
3	Not analyzed		47, XX, del(3)(q25.31q25.32),del(3)(q25.32q26.1),del(3)(q26.1),del(3)(q26.32), del(3)(q26.32q26.33),chromothripsis(4)(q22.1q34.1),del(5)(q21.3q22.2),del(5)(q35.1),del(5)(q35.1),del(5)(q35.1q35.2),del(5)(q35.3),+ 21
4	yes	yes	47, XX, dup(X)(p22.33p11.22),dup(7)(q11.21q36.3),del(17)(q25.3),+21
5	yes	yes	47, XY,+21
6	Not analyzed		50, X,i(X)(p22.33p11.1)x3,der(X)(q),del(9)(p13.2)/PAX5,+ 21

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
