# Peer review of "Copy Number Changes and Allele Distribution Patterns of Chromosome 21 in B Cell Precursor Acute Lymphoblastic Leukemia"

_cancers, 2021, doi:10.3390/cancers13184597_

Round 1
Reviewer 1 Report
The authors performed cytogenetic, FISH and array analyses in 578 ALL patients (including 26 with a constitutional trisomy 21) in ordert o evaluate numerical and structural abnormalities of chromosome 21. The authors performed a large amount of work and describe in detail the different ALL subgoups and their respecte absence or presence of different chromosome 21 abnormalities. However, it remains elusive what the real conclusions are and how these might impact on further research or on patient care.
Minor remarks:
The ALL subgroup that is named „Hyperdiploid, GW CN-LOH“ by the authors is usually referred to as low hypodiploid/near triploid – it is well known that the near triploid clone – which is infact a low tetraploid clone – arises from doubling oft he low hypodiploid clone – that is why it is obvious that those chromosomes which are lost in the low hypodiploid clone are CN-LOH in the near triploid clone.
Legend figure 3: change der(12)t(21;21) into der(12)t(12;21)
Author Response
We thank the reviewer for her/his comments. Our intention was to provide a complete and thorough overview about the entire spectrum of array-detectable chromosome 21 copy number alterations in childhood BCP ALL. Rather than doing so by purely reviewing the literature, we thought to base it on own cohort of array-analyzed patients and the insights we gained from the practical diagnostic experience. By doing so, we not only came across the usual well-known suspects but encountered various, originally difficult to explain patterns that we were eventually able to resolve and reinterpret by also taking into account the results of complimentary approaches, such as cytogenetic, FISH and STR-analyses. Cases in point are
- the unique case with a dic(9;20) and three chromosome 21 copies but an evenly retained balanced heterozygous allele pattern,
- the disclosure of meiotic recombination sites in hyperdiploid Down syndrome cases with somatic tetra- and pentasomies, a finding that, to the best of our knowledge, has hitherto neither explored nor documented in such detail and, finally,
- the cases with a GW CN-LOH.
In contrast to the reviewer's notion, these GW CN-LOH cases are not merely duplicated hypo- or near triploid clones. They have only between 48, 50, 52 or 54 chromosomes and, therefore, represent a distinct novel entity that can only be ascertained by array analyses. As pointed out, argued, and explained in considerable detail in reference 51 (which we updated sine it was published in the meantime), such cases do not require to go through consecutive steps that eventually duplicate an afore generated hyperhaploid genome but more likely derive from a single mitotic cell in a single step.
Being aware of the difficulties in interpreting especially all the various interrelated forms of hyperdiploidy, the intended purpose of our manuscript is therefore to provide a kind of guidance and to alert all those that deal with this diagnostic subject to the necessity to validate any unclear array data with FISH and/or any other applicable technologies. As some of our examples show, the detailed analysis of such cases can not only improve our understanding of the underlying causative mechanism and lead to a more polished delineation and subdivision of such entities that eventually may also indirectly have an influence of patient care.
To reinforce these points in our manuscript, we therefore added the following f sentences (indicated in red) into the revised version of our manuscript:
Lines 357-360: Low hyperdiploid genome-wide CN-LOH cases form a unique array-definable entity. They have only between 48 and up to 54 chromosomes and most likely derive in a single step from a single mitotic cell rather than from the genomic duplication of an afore generated hyperhaploid cell
Lines 453-458: The presence of an extra copy of chromosome 21 strongly influences the manifestation and maintenance of specific types of leukemias, in particular those with a CRLF2-overexpressing lesion and those with a dic(9;20). Whereas an inborn trisomy 21 primarily predisposes to the development of the former, it does not at all so in the later, although the acquisition of an extra copy of chromosome 21 is otherwise a very common attribute of both leukemia types in constitutional normal individuals.
Lines 463-464: By doing so, we were able to authenticate the preserved heterozygous allele pattern of a trisomy 21 in a dic(9;20) case and clarify ….
We thank the reviewer for alerting us to the transposed digits in Figure 3 and have corrected this mistake accordingly.
Reviewer 2 Report
The authors present the different chromosome 21 abnormalities that can be observed in childhood ALL. The description is interesting and detailed. The subject is not so new but the patient cohort is large and the analyses are precise and thorough.
The sentences in the introduction and some parts of the results could be simplified and shortened.
The authors explain that they used several studies that almost did not include any ETV6-RUNX1 positive cases. This results in a biased low number of chromosome 21 alterations in their series. However, it is precisely these anomalies that are studied. Could the authors explain why they used these studies?
The observed anomalies and the hypotheses on their occurrence are very detailed. However, there is no mention of the function of the studied genes, of the impact of these abnormalities, of the associated estimated prognosis or of the potential clinical impact. The fact that there is no context and no potential impact on the patient's management greatly limits the interest of the manuscript. It could be very interesting to have some data on the outcome of patients in terms of response to treatment and survival according to the anomalies described.
Author Response
We thank the reviewer for her/his comments. As we pointed out, our study is based on 578 patients that were studied partly retro- and partly prospectively in a prospective population-based manner as part of the still ongoing AIEOP-BFM ALL 2017 trial. Given that we only diagnose between 50 to 60 new ALL cases (including T ALL) per year in Austria, the analyzed ones were compiled over a long period and therefore also enrolled in different treatment studies. Apart from the in any case overall low number of cases, we also see only a comparatively low number of relapses. The reason, why ETV6-RUNX1 cases are underrepresented, results from the fact that, as in many other similar studies, these cases are usually excluded a priori from array analyses because of their overwhelmingly excellent treatment outcome. Secondary chromosome 21 abnormalities in these instances are therefore more of academic and research interest than of practical clinical value. The ETV6-RUNX1 cases documented herein, derive from those 304 cases that were diagnosed between May 2015 and the end of 2019, a period which comprised all consecutively diagnosed ALL cases.
Unfortunately, all these facts preclude any meaningful analysis of outcome data. However, as explained in more detail in the response to reviewer 1, it was also not our aim to provide such analyses.
Round 2
Reviewer 1 Report
no further comments
Author Response
Although the authors addressed all major comments raised by the reviewers, there are still some concerns regarding the impact of the study, patient cohort's characterization and use of genomic data as alternative or integrative method of analysis.
In particular:
Although the large amount of data, the conclusions still lack impact on further research or on diagnostics and treatment in the clinic. These should be more deeply discussed in the conclusions.
We have now added another short paragraph and a table that includes only recently obtained information about the association between trisomy 21 and the IKZF1 N159Y mutation (lines 370-382) and included this information also in the "Conclusion", whose first sentences we have modified. We sincerely hope that this novel information will now increase the value of our manuscript.
The authors used cytogenetic, FISH and array analyses to dissect chromosome 21 abnormalities, however genomic data are missing. With the increasing number of studies performing whole genome sequencing to identify genetic alterations in leukemia, it would be interesting to perform, in a subset of cases, a comparison between conventional data here generated and copy number and structural variations data from whole genome sequencing. May those data be available?
Unfortunately, we do not have any WGS data for comparison.
The resolution of the figures is very poor and some labels cannot be read.
We checked the submitted Word as well as PDF files and do not understand the editor's comment, since even when we expand the Figures, the resolution remains fine in the downloaded files. Moreover, we also had submitted separate TIFF files and can provide the original Photoshop files if that is deemed necessary.
Table 1 is a summary of the different subtypes included in the study, however a full list of all patients is missing. It would be helpful for the reader adding a Supplementary table including all patients analyzed in this study, with columns for clinical features, cytogenetic and molecular alterations (eg. known genetic alterations) and for results from this study according to each methodology.
We performed cytogenetic, FISH and array analyses in all 578 patients reported in this paper. It is extremely difficult for us to provide a complete list with all the requested data of these 578 patients. We believe that this request goes far beyond the scope of our intentions and would not add anything of particular value.
Can single cell-DNA sequencing technology studies help to dissect the clonal origin of these aberrations? Do the authors have data or can discuss it in the conclusion?
Single cell sequencing would certainly help to resolve the enigmatic combination in the case with the discrepant copy number and allele pattern. We have now added this statement.
Please be consistent in using the correct nomenclature for genes or proteins.
We took care to use the correct nomenclature throughout. We presume that your criticism refers to the different usage of CRLF2-expression. The italicized version refers to the RNA expression and the non- italicized one to the FACS-determined protein expression. In order to avoid potential misunderstandings and confusions we have now used the italicized version throughout the manuscript.
Cytogenetic and FISH analyses are not included in the “Methods section”.
As requested, we have included now the respective methods.
It is not clear from the text if “24 such B-other (including one with a matched relapse)” with ERG deletions are all DUX4 rearranged cases, if this is the case please re-phrase since DUX4-rearranged ALL is now an established B-ALL subtype. Characterization of ERG deletions lacks novelty. Please can the authors better comment on the relevance of the results from this study?
ERG-deletions were seen in 21 CD371-positive and RNAseq-verified DUX4-rearranged cases. The respective information was not available in one case, one of the two CD371-negative ones would have otherwise qualified as IKZF1plus and the other one had only a del(9)(p13.2) that encompassed the PAX5 gene. This statement has now been added.
The manuscript requires editing of English language and style.
Although we would be very happy to do so we would appreciate more precise instructions of what needs to be changed.
For easy reference, we have marked this time all the additions and changes in the second revision of our manuscript in blue.